# Trends and Opportunities of Tertiary Education in Safety Engineering Moving towards Safety 4.0

Vendula Laciok [1], Katerina Sikorova [1], Bruno Fabiano [2] and Ales Bernatik [1,*]

1    Faculty of Safety Engineering, VSB-Technical University of Ostrava, 708 00 Ostrava, Czech Republic;
     vendula.laciok@vsb.cz (V.L.); katerina.sikorova@vsb.cz (K.S.)
2    DICCA-Civil, Chemical and Environmental Engineering Department, Polytechnic School, Genoa University,
     16145 Genoa, Italy; brown@unige.it
*    Correspondence: ales.bernatik@vsb.cz; Tel.: +420-597-322-833

**Abstract:** Industry and related work and workplaces are constantly changing as a result of the implementation of new technologies, substances and work processes, changes in the composition of the workforce and the labor market, and new forms of employment and work organization. The implementation of new technologies represents certain ambivalence. Next to the positive impact on workers' health, new risks and challenges can arise in the area of process and occupational safety and health of people at work. On these bases, it follows the need for predicting and handling the new risks, in order to ensure safe and healthy workplaces in the future. The aim of most forecasting studies is not only to identify new emerging risks, but also to foresee changes that could affect occupational safety and health. However, a number of questions still require proper investigation, i.e., "What impact do new emerging risks have on tertiary education in the area of Safety engineering? Has tertiary education already reacted to progress in science and research and does it have these innovations in its syllabus? How are tertiary graduates prepared for the real world of new technologies?" This paper represents a first attempt in the literature to provide answers to the raised questions, by a survey approach involving academics, Health Safety and Environment (HSE) industrial experts and university students in the Czech Republic. Even if statistical evaluation is limited to a single Country and to a small sample size, the obtained results allow suggesting practical recommendations that can contribute to ensuring new challenges in the area of education by addressing relevant culture issues needed to support new workplace realities according to the newly defined Safety 4.0.

**Keywords:** digital technologies and risk; occupational safety and health; process safety engineering; Safety 4.0; sustainable education



## 1. Introduction

The developmental process in the management of manufacturing, processing, and chain production was quite recently defined by Schwab [1] as "Industry 4.0", while referring to the convergence of manufacturing and processing with the digital revolution, artificial intelligence, the Internet of things, and with every smart device. In this regard, companies are entering into a new phase of development. Artificial intelligence, Big Data, 3D printing, 5G Internet, Internet of things (IoT), and nanotechnology are often mentioned among the more specific technological trends in production and services with an impact on human work [2–5]. Innovation leads to new processes, under yet unknown conditions and introduce potential new hazards from novel or emerging technologies and materials with a continuous need for further research and development [6]. Technological trends and digitization (Industry 4.0) are changing the nature of work in terms of the number of jobs in different sectors, the forms of working relationships, and the content of the activity performed.

Between 2000 and 2015, the use of information technology in the EU increased by 16%, thus, representing a convincing example of the rapid digitization of the work-

place, with which it is difficult to keep up without the appropriate skills [7]. Starting from Germany, the novel trend rapidly expanded to industrialized countries worldwide, with the development of specific actions for investment and research lines [8]. Analyzing the current situation on the labor market, it is evident that the approach towards Industry 4.0 differs not only at a national level, but also in individual production sectors, both in manufacturing and process companies. In fact, each country relies on different resources, finances, working groups, and legislative frameworks, but at the same time, there is a system of cooperation and projects to serve as motivation for moving on in the world of robots, digitization, and automation. Not surprisingly, the European Council identified market digital transformation as a pillar of EU recovery in reaction to the various waves of Covid pandemic [9], by proper funding actions on [10]:

- fostering next generation of digital technologies;
- developing capacities in strategic digital value chains;
- upgrading digital capacities in education systems;
- enhancing the EU's ability to protect itself against cyber threats and to reach environmental goals using digital technologies.

Even if each Country represents and defines selected progress steps and goals in Industry 4.0, implementing initiatives and national programs [11], it should be remarked that a sustainable modification of an industrial system implies a thorough discussion and analysis of novel hazards and the potential effects on worker health and safety. A major issue in facing Industry 4.0 businesses is connected to face-to-face, or virtual training, of existing workers and to recruiting new workers who are better equipped to learn [12] novel technical skills and solve new problems. Co-evolution of manufacturing philosophy and the approach to occupational health and safety (OHS) is ongoing: the implementation of new industrial concepts based on information decentralizing and decision-making requires, correspondingly, the evaluation of the OHS consequences (positive and negative) of this industrial revolution [5], as well as the elaboration of innovative safety metrics including leadership, training, and onboarding. Furthermore, when considering higher education, effective course design should develop understanding and competences in the fundamental process safety principles and practices allowing effective risk management, but these tasks are nontrivial and often rare across universities, especially when considering novel requirements connected to Industry 4.0.

The reminder of this article is presented as follows. Section 2 introduces the novel concept of Safety 4.0 and illustrates emerging risks. Section 3 outlines the conceptual framework that informed the research introducing the methodology and gap investigation method based on a questionnaire survey and data collection strategy in a selected country. Section 4 identifies the key research issues regarding new technologies, connected hazards, and higher education gaps. Section 5 presents the results from questionnaire on the series of selected key themes, while Section 6 discusses the actual implication of the results. Conclusions are drawn in Section 7, including the current limitations of research in this area and in this study.

## 2. Safety 4.0: Why and What

The ambivalent attitude toward new technologies stems from the conflict of negative and positive expected impacts and technological advances in industrial activities can give rise to improvement in productivity and in occupational health and safety, but not necessarily simultaneously [13].

Both technology itself and the way it is used can represent a risk factor. First of all, technology or some of its effects can be unreliable, despite the best intentions of the users. Secondly, there are consequences in which technology as such does not fail, but directly engages the user in achieving an unethical or otherwise negative goal, or achieves a positive goal with known negative side effects of its use. A comprehensive example of immediate readability regarding the paradigm of industry 4.0 evolution is provided by Keller [14],

illustrating by simple numerical indicators the increasing importance of digitalization comparing their relevance in 2018 and 2025, as depicted in Figure 1.

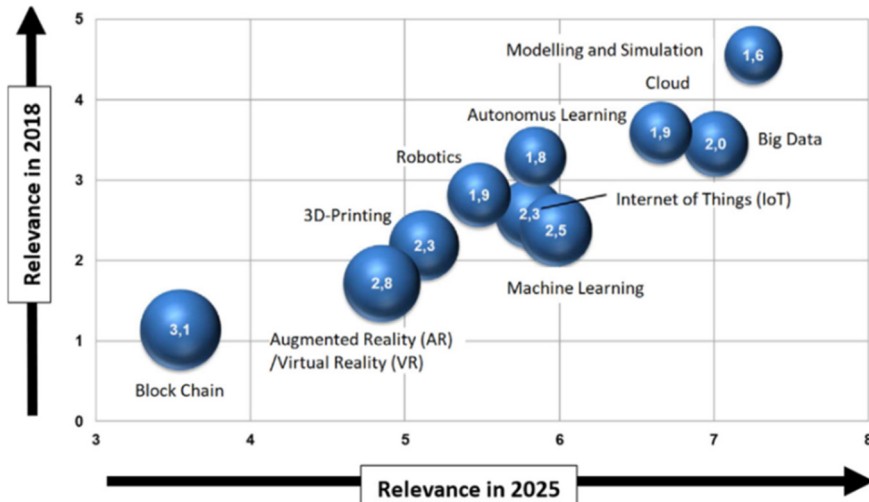

**Figure 1.** Industry 4.0 evolutive trend (from [14]).

However, technology itself exerts its effects on people long before it is actually deployed and, unfortunately, long after that. The current fast pace of technological innovation implies that the ability to learn from past experience is limited. New digital equipment ranges from drones and robots to wearable for workers, proximity sensors for vehicles, and even smart personal protective equipment (PPE), which should be selected starting from the actual analyses of accident and near miss [15] to prioritize the adoption of new technologies by proper lagging and leading indicators metrics. Accordingly, new approaches are needed to re-engineering process safety and the way technological risk should be managed towards the newly defined Safety 4.0, introduced at the 16th EFCE Loss Prevention Symposium held in Delft, The Netherlands [16].

This neologistic term is not only limited to "computational safety science as a new paradigm for safety science in the era of big data and Industry 4.0" [17], but includes a full methodological and conceptual transition and a new series of paradigm shifts towards the novel safety concept. From one side, we need understanding of the potential effects of technological change on employment, worker individual well-being and health, performance and impact at the organization level, identifying strategies to increase trust and acceptance for the use of disruptive digital technologies. From the other side, implications for health and safety connected to the technologies driving Industry 4.0, in terms of work organization, OHS regulatory and legislative framework, OHS management systems, and occupational risk management systems needs need a careful analysis [18].

On these grounds, Safety 4.0 is here defined as "a proactive shift to the science of process and occupational safety, which focuses on system resilience and dynamic risk management and is based on four pillars, i.e.,: interoperability (IoT), information transparency (digital twins), technical assistance and decentralized decisions".

The above mentioned four guidewords are illustrated in the seminal paper by Herman et al., to which the reader is addressed for further details [19].

Modifications of process and personal safety connected to Safety 4.0. will be necessary in the near future including, e.g., automation of hazard identification studies and risk assessments, advanced and augmented reality use for safety training of operators, implementation of digitalization in all safety management activities (pre start-up safety review, work permits, standards, safety tours emergency simulations, etc.). The use of improved digital tools to conduct process hazard analysis has the potential to address some of the recognized shortcomings of the traditional HazOp process, such as lack of

completeness in identifying initiating causes and scenarios and inadequate propagation of deviations [20].

Figure 2 provides a picture of traditional process safety domains, from which it is evident how risk analysis is the most important part of the interest areas. They include topics relevant to "substance properties", e.g., combustion, thermal stability, deflagration, detonation, toxicity, test methods, criteria, classification, and regulation. The other domain, "system safety" includes safe design principles, Fault Tree Analysis, and reliability engineering. "process technology engineering" includes a wide range of topics, e.g., operation and organization: Inherent safety, Safety Management Systems etc. The core layer "risk analysis" includes a wide range of established and more recent tools and concepts, e.g.,: Layer Of Protection Analysis, Consequence analysis, QRA, Decision analysis, Criteria, natural events triggering technological disasters (NaTech events), Domino effects. The main drivers for the evolution towards the previously defined Safety 4.0 are highlighted in the already mentioned Figure 2.

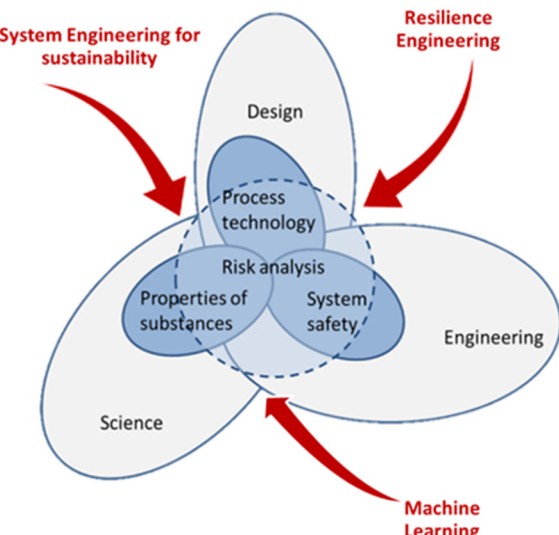

**Figure 2.** Traditional process safety areas and evolution required by the transition towards Safety 4.0 (modified from MKOPSC [21]).

Here, "system engineering for sustainability" comprises full development of system thinking approaches, such as Leveson approach System-Theoretic Accident Model and Processes (STAMP), STAMP Analysis (STPA), and Bayesian Belief Net (BBN) modelling. Issues relevant to "Machine Learning" (ML) and "Resiliency", which are interconnected, include, e.g., data-driven approaches, risk/process conceptual design, resiliency, flexibility, operability, cluster-driven design, dynamic risk, modular plant development, cross-fertilization, and spin off of techniques. In particular, resilience analysis by digital technologies, covering both plant and management system level, should hedge for the unexpected and unknown, thus, contributing to loss prevention, business continuity, and sustainability. Correspondingly, a multilayer framework is required in risk assessment (see Figure 3), starting from the inner layer, i.e., process plant engineering integrating technologies such as digital twins, hybrid-modeling (first principles and data-driven techniques), and novel sensors.

It is noteworthy evidencing that many challenges are still to be addressed in order to make further progress in safety and sustainability of plants and processes moving towards Safety 4.0. As commented in the extensive review [22], safety and sustainability are closely linked and share the same pillars, i.e., economy, environment and society, with the former offering an operational command on the latter. From one side, emphasis in digitalization is about identifying the benefits of this technology to the management of major hazards, from the other side is connected to the need of ensuring the risks of the technology are

understood and correctly managed [16], for example in relation with loss of control over installation, i.e., "digital runaway" and lack of understanding of the process.

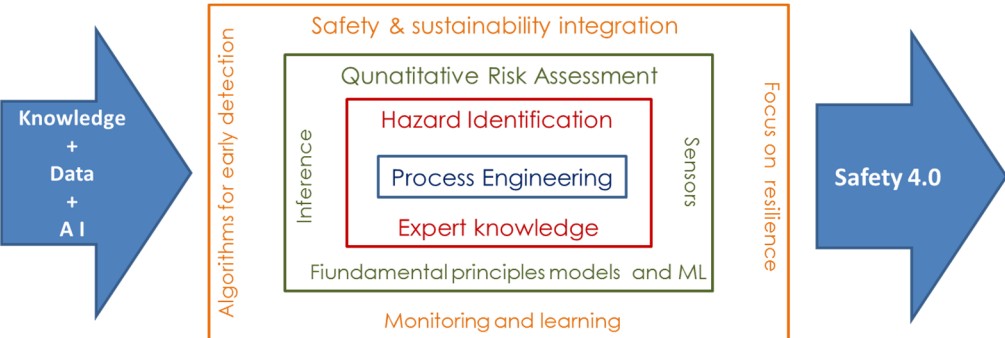

**Figure 3.** Multilayered approach improving traditional risk assessment towards process resiliency and Safety 4.0.

In fact, despite the improvements and new developments leading to process industry 4.0, accidents still happen as present methods do not guarantee perfect hazard identification and skip turn-arounds and other abnormal situations; risk assessment is often only qualitative and it is fallible, even when it is quantitative. Plants are becoming more complex due to additional requirements of new designs and technology with more automation, and novel hazards are introduced by digitalization so that in due time a further revised version of Seveso Directive to cover emerging issues may be needed [23], e.g., considering innovative safety barriers and technical measures for equally protecting human health and the environment [24]. As anticipated, the European Commission favors digitalization and EU leaders agreed that at least 20% of the funds under the Recovery and Resilience Facility will be made available for the digital transition, including for small and medium-sized enterprises (SMEs) [10]. Core strategic objectives for Europe include the creation of a credible and efficient technical infrastructure and the reduction of digital barriers and the digital divide. Determining factors for the development of the digital society [25] are human capital, genuine use of the Internet, digital integration, digitalization of the economy, promotion of citizens' digital skills, and upgrade of digital capacities into the education system.

### 3. Materials and Methodology

This work focuses on occupational risks of Industry 4.0 and the analysis of the current situation on the labor market from the perspective of employers and requirements for higher education, considering as relevant case-study the Czech Republic universities. The proposed methodology is schematically depicted in Figure 4.

Similarly to the approach by Hoła and Nowobilski [26], on grounds of expert opinions and extensive literature review, it was possible defining the basic factors, challenges, and hazards that characterize Industry 4.0. Analysis of subject literature and issues related to occupational safety evidences that this topic is still scarcely explored. The focus of the survey is to analyze the current situation on the labor market from the employers' perspective and the requirements they have from students, i.e., graduates of schools in the Czech Republic. The analysis is aimed at employees who work as a control authority in the Czech Republic in the field of occupational safety and processes and who work as representatives of employees in the field of Occupational Safety and Health (OSH) and trade unions in companies. The analysis includes as well university students performing a pilot application at VSB—Technical University of Ostrava (VSB-TUO), Faculty of Safety Engineering, in terms of their knowledge of Industry 4.0 and connected safety issues. A quantitative approach of public opinion research was used for the pilot analysis, i.e., a questionnaire survey using statistical techniques. Globally, the different questionnaires involved both academics, i.e., scholars in the field of process and occupational safety and practitioners, such as members of control authorities and facility technical managers. All participants are

characterized by a high degree of substantive experience and heterogeneous backgrounds, useful to cover all relevant aspects of the matter under investigation.

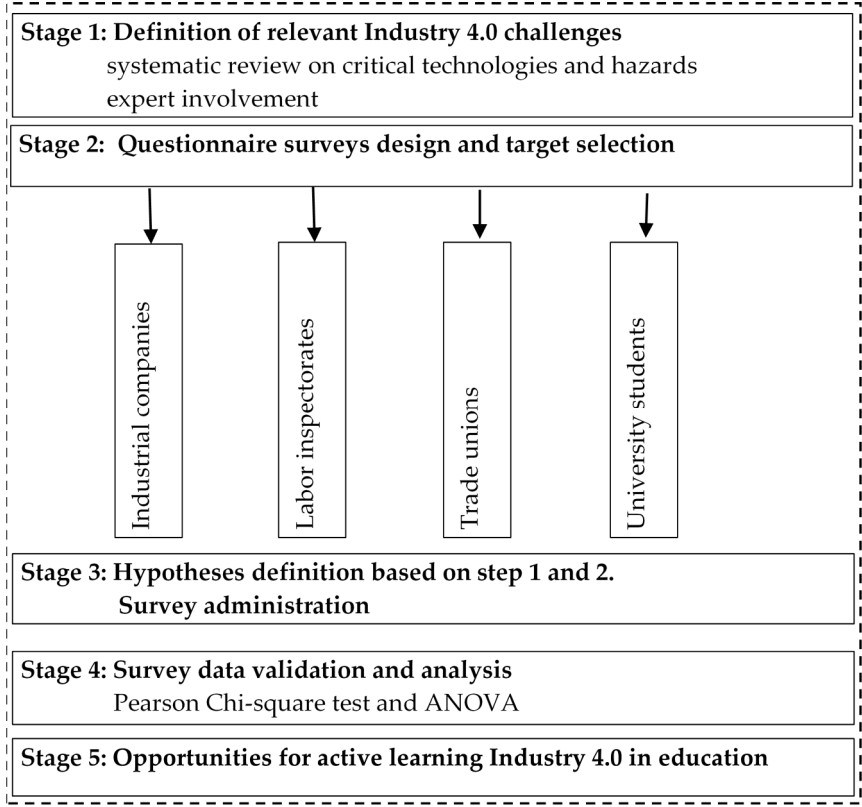

**Figure 4.** Developed multistep methodology for pilot analysis.

The questionnaire was designed by consultation with three academic advisors along with two industrial experts from HSE departments in order to gain a deep understanding of professional needs. The questionnaire design followed in part the type employed in another study on safety culture and climate [27] and was preliminarily validated by five consulted experts. Due to GDPR, the survey was anonymous. Analysis method included frequency distribution, chi square test, and one-way analysis of variance of questionnaire data, upon preliminary validation.

The data, obtained by means of a questionnaire survey, were evaluated using three hypotheses tested at the 5% level of significance ($\alpha = 0.05$). A goodness of fit test (Pearson Chi-square test) was used to test the hypotheses, which is used for comparing frequencies in categorical data.

The general formulation of the null and alternative hypothesis as summarized as follows:

- $H_0$: frequencies equal
- $H_A$: frequencies differ significantly.

Test statistics were calculated according to:

$$\mathcal{X}^2 = \sum_{i=1}^{n} \frac{(O_i - E_i)^2}{E_i} \tag{1}$$

where:

$O_i$ = the observed number in the $i$-th group,
$E_i$ = the expected number in the $i$-th group,
$n$ = number of groups,
$\chi^2$ = value of test statistics chi-square distribution.

The value of the test statistics was converted to the significance level (*p*-value) with CHIDIST and subsequently compared with the selected significance level (*α*). The null hypothesis was rejected in favor of the alternative hypothesis if the attained materiality level was lower than the selected materiality level. Statistical elaborations were performed customizing evaluation sheets by Microsoft Excel 2010.

According to the outlined methodology, in the next section we will present a mini review to identify industry 4.0 and digital technologies for which new education and continuous learning is needed, to promote their ethical safe and responsible use in the process industries.

## 4. Survey on Relevant Industry 4.0 Technologies

Advanced technologies in the form of devices and technological units tend to have the most significant impact. Table 1 summarizes an overview of selected technologies according to their occurrence in individual key studies. These are the identified most important technologies with a significant impact on technology areas and the economy within the next 10 years [28–30] and are considered relevant for the applicative phase of this paper. They are selected as pertinent examples of potential unknown accident scenarios in case of hazardous deviation, due to epistemic uncertainty regarding failure modes that consequently cannot be quantified and included into a detailed risk description.

**Table 1.** Selected technologies overview [28–30].

| Technologies | |
| --- | --- |
| Artificial Intelligence | Cloud Computing |
| Block chain | Smart electrical network |
| Internet of Things | Communication between machines |
| Augmented reality | Advanced production |
| Virtual reality | Interface human-machine |
| Robotics | Advanced storage energy |
| 3D printing | Nanomaterials |
| Drones | Nanotechnologies |
| Big data | Autonomous automobiles |
| Cyber security | Advanced production |

The following subsections will focus on the potential risk sources and significant hazards associated with the use of given new technologies that will be adopted for proper questionnaire design. Generally speaking, the technologies driving Industry 4.0 bring new challenges for productivity increase by the integration of digital systems of production with analysis and communication of all data generated within an intelligent environment, but imply modifications within the broad area of Safety 4.0, and related management systems. Even though significant, cyber security and autonomous vehicles are outside the purpose of this research paper.

### 4.1. Smart Electrical Network—5G Internet

The fifth generation of mobile networks (5G) will be the next step in the development of telecommunications standards, far beyond what the current 4G/LTE networks offer. 5G technology will meet the needs arising from the ever-increasing demands of mobile network users and the needs of those who want to implement innovative concepts (autonomous vehicles, smart factories—predictive maintenance, or smart cities) [31]. Compared to existing 5G mobile networks, it will be able to support many more devices with much higher data rates, extremely low latency, and very high reliability. In this way, 5G technology will provide users with high quality services, enabling as well highly reliable and massive communication between devices.

First, 5G networks will use frequencies around the 3.5 GHz spectrum band. This band is similar to that used in existing networks (see Figure 5). Later, the network introduces frequencies of 24 GHz and higher, sometimes referred to as mmWave (millimeter wave).

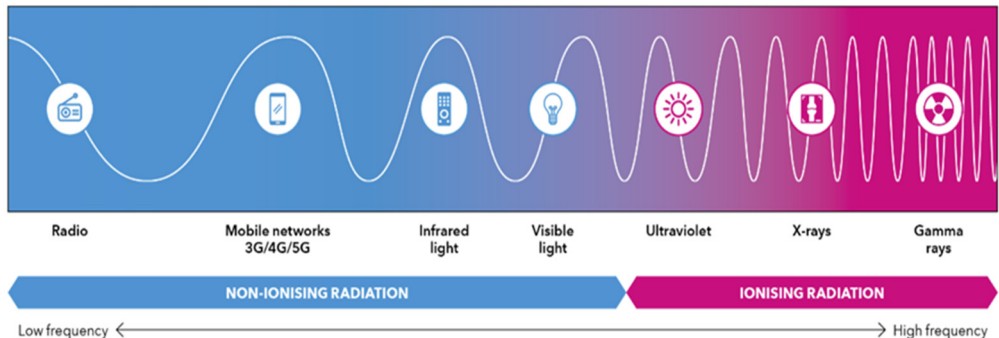

**Figure 5.** Electromagnetic radiation [32].

The potential adverse effects of 5G technology stem from the radiation itself and its interaction with tissue and target structures. 4G network technology was mainly associated with carrier frequencies in the range of 1–2.5 GHz (mobile phones, WiFi). The wavelength of radiation at 1 GHz is 30 cm and the depth of penetration into human tissue is several centimeters. In its highest power (high-bandwidth) mode, 5G network technology is mainly associated with carrier frequencies at least an order of magnitude greater than 4G. The depths of penetration into human tissue with 5G technology will be of the order of a few millimeters [33].

If there are adverse effects resulting from 5G technology, the major adverse effects will be manifested on the skin surface (skin cancer, cataract, and other skin conditions). Russell [34] claims that biological reactions to millimeter wave irradiation can cause adverse dermal effects that can lead to physiological effects on the nervous system, heart, and immune system.

With its diverse broadband frequency and a dense network of transmitters, 5G significantly increases exposure to electromagnetic radiation. There are significant data gaps in millimeter wave research (mmWave) and mixed frequencies on biological effects and long-term exposure in vulnerable populations (children, pregnant women, etc.) [35,36].

### 4.2. Nanotechnologies/Nanomaterials

The origins of nanotechnologies go back to the development of electron microscopy and X-ray diffraction methods that have made it possible to determine the structure of materials at the atomic and molecular level, to understand its impact on material properties and to manipulate it to create new materials—nanomaterials (NMs) with unique properties [37].

Nanomaterial, according to the Recommendation of the European Commission (2011/696/EU), is defined as "a natural material, by-product or material produced containing particles in an uncomplexed state or as an aggregate or agglomerate in which 50% or more of the particles have a size distribution one or more outside dimensions in the size range 1–100 nm".

For particles smaller than 100 nm two terms are used. The term "nanosized particle" is used when referring to so-called "purpose-made" nanoparticles—engineering nanoparticles, and the term "ultrafine particle" (UFP) includes surrounding non-purpose-produced nanoparticles [38]. Ultrafine particles have been present in nature since time immemorial (mountain air contains $10^3/cm^3$ particles). Only new instrumentation and new approaches have allowed the specification of particles smaller than 100 nm. When particles of a given material are below 100 nm, the physical-chemical properties of the surface begin to predominate over the properties of the material, and the particles begin to behave as if they were entirely surface-based. One of the most significant phenomena of this process is a strong increase in chemical reactivity, which may result in a change in toxicity [39].

Nanomaterials/nanoparticles/UFPs and their applications can make a significant contribution to improving the quality of life and addressing the major challenges facing human society. However, the same properties that make NMs/UFP/nanoparticles unique

and desirable may also be undesirable in terms of health effects or physical hazards (ignition, explosion, etc.).

Cells of the human body are a hundred thousand times larger than nanoparticles. When nanoparticles enter the lungs by breathing, they are able to overcome biological barriers and not only through the bloodstream to reach various places in the human body. The most common exposure to nanoparticles is through the airways and, thus, the lung is the least effective barrier. If nanoparticles penetrate into the bloodstream, they can, like substance molecules, gradually penetrate into the cells and deposit in some specific organs that are most supplied with blood (heart, liver, etc.). We can also expect the most serious damage in these organs. In addition to blood circulation, nanoparticles penetrate relatively easily through the nasal membrane from where they are transported through the olfactory and cranial nerves to the central nervous system [40,41].

The behavior of nanoparticles varies, because it depends on the material they are made from, the size, and the behavior of the nanoparticles in the body of exposed persons cannot be predicted. Thus, most of the available toxicological data are based on narrow-focus studies conducted on cell cultures or animals and are difficult to extrapolate in humans. To date, the databases about nano-toxicology have a limited amount of information to establish suitable and widely accepted workplace safety regulations, both in relation with the toxicological/hazardous properties, lifecycle assessment, potential exposure pathways, and the actual estimation of the number of workers exposed. The toxicological effects of nanoparticles on the cell are neither entirely clear, nor fully experimentally investigated [40,42].

Nano-toxicology databases have a limited amount of information to establish suitable and widely accepted workplace safety regulations, both in relation with the toxicological/hazardous properties, lifecycle assessment, potential exposure pathways, and the actual estimation of the number of workers exposed [43].

An example of self-defensive organism reaction consists in transporting NPs from the respiratory tract according to the clearance process. Clearance consists of two phases, namely:

- rapid—in the tracheobronchial area of the lungs, where the mucus expels the particles out of the respiratory tract;
- slow—absorption of particles (phagocytosis) [41].

Currently, the consequences of possible hazardous human interaction with novel nanotechnologies (likewise 5G technology) are not completely assessed, so that the precautionary principle application represents a need and a must. Using the sustainability framing shows the path for process production development with wider application of inherent safety guidewords [44] and avoidance of chemical reaction wastes, addressing cleaner and sustainable production [45].

### 4.3. Additive Production—3D Printing

According to OSHA-EU, the main hazard connected to additive production is the generation of dust, which may have the same negative impact as nanoparticles/nanomaterials when workers get exposed to them, as previously briefly discussed. Exposure to ultrafine 3D printing particles has gradually become an emerging occupational and environmental health interest. 3D printers have been widely used as a tool for prototype manufacturing in industries, schools, and homes. During operation, 3D printers emit ultrafine particles and volatile organic compounds that could be inhaled by the user and induce adverse health effects [46,47]. The inhalation and deposition of such particles in the human respiratory tract could lead to various adverse health effects such as asthma [48] and respiratory inflammation [49].

Depositions of ultrafine 3D printing particles in the tracheobronchial airways were found to be 1% to 13% depending on the particle size [41]. Inhalation of some volatile organic compounds has been associated with adverse health effects, including irritation of

the respiratory system, sensory effects and mutagenic effect [47] and their minimization requires the development of new solvent free routes [50].

### 4.4. Drones

The name drone comes from the English word "drone". In the Oxford Dictionary [51], in correspondence to this word, we find expressions such as drone, prolonged low tone, growl, a man who is not beneficial to society and lives at the expense of others, and ultimately also a remote-controlled unmanned aircraft or missile.

Drones are currently used for commercial purposes, public security, as a hobby, and for scientific research. Short-term and long-term predictions show the widespread use of Unmanned Aerial Vehicles (UAVs) in various sectors. According to the Federal Aviation Administration, these are sectors such as insurance (5%), emergency management (8%), agriculture (21%), construction, industrial and utility inspection (26%), real estate (26%), and aerial photography (34%) [52]. Main challenges are connected to human error reduction, travelling time, and environmental impact reduction by reducing traffic jamming, possibility of reaching remote locations unconnected by standard transportation means.

Despite cognitive, consequentialist, and "rational" risk-assessment theories, AV risk is still a challenging and important future direction for research also in view of social acceptability [53]. However, the technology highly relies on complex system automation vulnerable to safety concerns, while in case of low altitude operations additional concerns include human and industrial-civil infrastructure safety at the ground.

There are many areas in which a drone can pose a risk by harming property and people: e.g., it can be misused for malicious intent as an explosive and flammable material or worse a hazardous chemical carrier, or employed as a guided weapon (pilotless kamikaze missions), or can interact with other civil aircraft flight paths (attack swarms). Additional hazards connected to drone misuse include illegal load delivery (smuggling) and opportunistic observation or spying of people leading the privacy of an individual or the protection of business secrets. In this regard, it is acknowledged the need of a careful consideration of drone controls adequacy over the impacts on two important values, namely public safety and behavioral privacy [54].

### 4.5. Robotics/Human-Machine Interface

Today's approach to industrial robotics is that the human is replaced by a robot mainly in non-ergonomic or inherent hazardous activities. Notable examples include manipulation of heavy objects or activities in areas unsafe for humans, performing dangerous tasks such as manipulation of toxic or hot objects, and additional potential high-risk deviations in the plant can be stopped automatically by robots and hazardous interventions can be performed without the physical presence of humans. Additionally, robots efficiently replace humans in operations that are monotonous and often repetitive or require a high level of accuracy.

Collaborative robots are within the Industry 4.0 concept a current trend in both industrial and service robotics. The aim of this new technology is to enable direct safe cooperation between the robot and the operator (human). Between manual production and complete automation, there is a grey zone where man comes into contact with machine. For a long time, the utilization of this area has been limited. For the safety of the operator (human), the machine could only be operated when the person was outside the working area. The main reasons for the use of collaborative robots include limited installation space for the planned workplace, improved ergonomics of the production process, frequent change of robot trajectory, improved product quality, or elimination of monotonous operations performed by the operator.

In most collaborative robot applications, "light robots" are used, as summarized in Table 2 where we identified some promising challenges for implementing Industry 4.0. It is easier and safer to work with these robots because they are not, by their very nature, able to

exert a great force. They have built-in direct drive motors with torque sensors that allow much higher sensitivity in collision detection [55].

**Table 2.** Overview of some collaborative robots (derived from firms' websites).

| Company | Name | Payload | Reach | Weight |
|---------|------|---------|-------|--------|
| ABB | YuMi | 0.5 kg × 2 | 500 mm | 38 kg |
| ABB | Roberta | 4/8/12 kg | 600/800/1200 mm | 14.5/19.5/30.5 kg |
| BOSH | APAS | 4 kg | 911 mm | 230 kg |
| FANUC | CR-35iA | 35 kg | 1813 mm | 990 kg |
| FANUC | CR-4iA | 4 kg | 550 mm | 48 kg |
| FANUC | CR-7iA (L) | 7 kg | 911 mm | 55 kg |
| KAWASAKI | duAro | 2 kg × 2 | 760 mm | 145 kg |
| KUKA | LBR iiwa 7 | 7 kg | 800 mm | 23.9 kg |
| KUKA | LBR iiwa 14 | 14 kg | 820 mm | 29.9 kg |
| RETHINK ROBOTICS | Sawyer | 4 kg | 1260 mm | 19 kg |
| RETHINK ROBOTICS | Baxner | 2.3 kg | 1041 mm | 75 kg |
| UNIVERSAL ROBOTS | UR3 | 3 kg | 500 mm | 11 kg |
| UNIVERSAL ROBOTS | UR5 | 5 kg | 850 mm | 18.4 kg |
| UNIVERSAL ROBOTS | UR10 | 10 kg | 1300 mm | 28.9 kg |
| UNIVERSAL ROBOTS | UR16 | 16 kg | 900 mm | 33.1 kg |

This drawback limits the original use of robotic capabilities and in particular their high performance (performing demanding tasks: lifting, long-term holding, working in hard-to-reach areas, etc.). Industrial robots are able to offer the required forces and performance, but are forced to work behind safety barriers (fences), as no tools are available to ensure safety. Therefore, the ergonomic advantages of industrial robots are not being fully utilized.

Robots are perceived as inter-connected physical objects equipped with sensors, actuators and controllers connected by internet for real time data collection and joint collaboration and from the safety perspective, there is the need to determine the potential environments in which they operate and the movement control measures under given conditions and plant areas [56]. The design of workplaces with collaborative robots is a constantly developing area of research with a minimum number of reference workplaces, which were created in non-laboratory conditions. Subsequent research areas should focus on the effective detection of the operator in the robot's workspace and subsequent identification of the possibility of collision and reaction to this situation, as well as the possible hazardous interaction of the operator with the workplace, possibly by real-time image acquisition of load/obstacle and elaboration by a dynamic risk assessment model [57].

## 5. Results

Mařík [28] draws attention to the fact that the fourth Industrial Revolution is more a fundamental change in people's thinking rather than a modification in technology. A number of new education requirements are placed on universities, which will in turn have to change the content and style of teaching at the burden of additional economic investments. These will move towards education, which will demand a change in the way students are evaluated and managed.

This research should be regarded as a first step in the analysis to ensure that graduates develop capabilities for the use of novel tools, by acquiring the required knowledge and skills for employment within the context of Industry 4.0. In the next section, we will report the numerical results obtained from the survey, on the selected case study. Recognizing that the relative novelty of the topic limited the possibility to enroll expert and respondents, it must be noted that the relatively small number of participants might have influenced the results of the study, thus requiring further validation.

### 5.1. Questionnaire Survey of Companies in the Czech Republic

The first questionnaire, consisting of 11 questions, dealt with the research of business entities in terms of Industry 4.0 in the Czech Republic. The aim of the questionnaire was to analyze the current state of companies, especially their position in relation to schools and requirements from graduates. The target group was companies that have implemented Industry 4.0 technology, so that their feedback would contribute to the future transformation of fields of study, in reaction to 4.0 Industrial Revolution. The total number of questionnaires evaluated was 87, of which 56 were completed in companies with fully established Industry 4.0 technology.

The proportion of respondents by company size was relatively even. The largest proportion of respondents (52) was represented by business entities within the industry, i.e., secondary sector. The following score was the tertiary sector with 33 respondents representing services—from transport, health, trade, education, culture and communal services, science and research, and high-tech technologies. The primary sector was represented minimally, with only two cases.

Out of 87 responses to the questionnaire survey, more than half of the respondents had Industry 4.0 technology and tools actually implemented at different levels within the company. Figure 6 shows the most frequently implemented technologies.

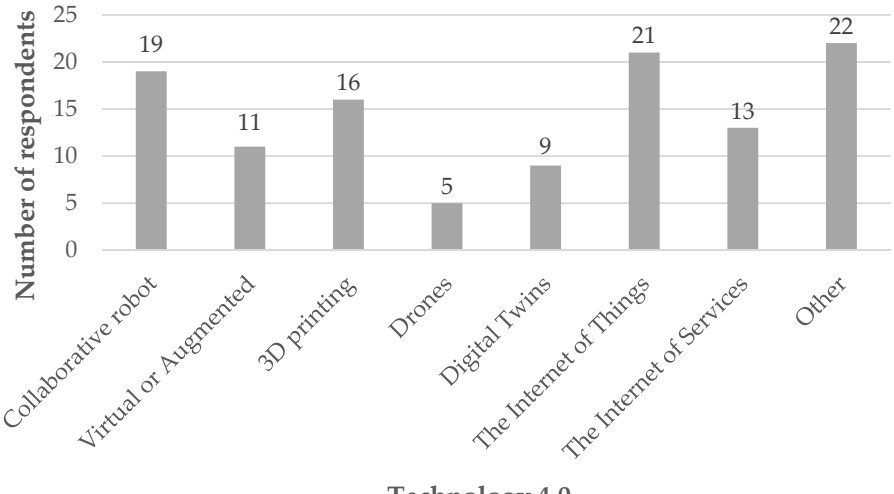

**Figure 6.** Representation of Industry 4.0 respondents.

A striking high percentage of respondents, i.e., 82%, explicitly prefer graduates with adequate study backgrounds covering the pertinent industrial field. A total of 74% of respondents were/are cooperating with a university, by offering internships to students during their studies. Many companies offer paid internships, but not as a mandatory rule, to gradually educating and training their future employees. The focus for the student is to acquire on-site practical experience, as well as to get more acquainted with the potential future employment market. Globally, 23 firms corresponding to 26% of respondents, reported no past university cooperation, with 15 companies open to active cooperation in the near future.

Seventy-four percent of respondents explicitly preferred graduates with work experience, even if a discrepancy between study and practical application is not seldom evidenced. There are students who have been actively seeking work experience in the field since becoming bachelor students and it is easier for them to gain valuable experience. Other students, however, who focus only on education during their studies, gain practical experience directly on-the-job, possibly by training. In several instances, contradictory business job offers are reported, i.e., the job is "suitable for graduates", while requiring "at least two years of experience". One solution to this issue is to promote higher commitment and motivation to expand cooperation between companies and universities.

A quite similar statistical figure regards the graduates' knowledge and skills: 53% of the respondents replied that the knowledge/skills of the graduates were adequate, while the remaining 47% reported a lack of experience in different knowledge domains. These areas were analyzed and divided into subject groups. The first one consists of knowledge that students could be developing in the future in their field of study: knowledge of legislation (national occupational and process safety, Seveso, ATEX, and PED Directives, etc.), mechanics, physics and materials, language skills, practice, diligence, new learning motivation, independence, analytical thinking. The second group consists of knowledge concerning selected fields of study. There is a demand for students of IT technology, electrical engineering, and mechanical engineering. The answers to the last question suggest that companies most often are keen to invest in 3D printing, augmented and virtual reality, as well as in collaborative robotics.

Three hypotheses were elaborated for this section of study:

1.  *Employers do not prefer graduates from a particular school.*
2.  *Students' knowledge is insufficient after graduation, in terms of employers' requirements.*
3.  *More than half of the interviewed employers who did not cooperate with universities by offering student internships would welcome future cooperation.*

**Hypothesis 1.** *To test this hypothesis, the answers to the question were as follows: "Do you prefer candidates that graduated from a particular school?" The response rates are summarized in Table 3.*

**Table 3.** Measured and expected response rates of the 1st hypothesis.

|  | **Measured** | **Expected** |
|---|---|---|
| Yes | 19 | 43.5 |
| No | 68 | 43.5 |
| TOTAL | 87 | 87 |

Since the obtained *p*-value ($1.49 \times 10^{-7}$) was lower than the significance level of 0.05, the tested hypothesis "Employers do not prefer graduates from a particular school" could be confirmed at the 5% significance level.

**Hypothesis 2.** *The answers to the following question were used to test this hypothesis: "Did you ever register a student lacked knowledge in terms of the technology you are using?" The responses are summarized in Table 4.*

**Table 4.** Measured and expected response rates of the 2nd hypothesis.

|  | **Measured** | **Expected** |
|---|---|---|
| Sufficient | 54 | 43.5 |
| Insufficient | 33 | 43.5 |
| TOTAL | 87 | 87 |

Level of significance achieved (*p*-value 0.0243) was less than the selected significance level (0.05). Just by looking at the data, it is clear that the hypothesis, "Students' knowledge is insufficient after graduation, in terms of employers' requirements", was rejected at a 5% significance level.

**Hypothesis 3.** *The answers to the following question were used to test this hypothesis: "Have you ever cooperated with universities by offering internships to students?" and "If not, would you be interested in such a cooperation?" The responses of respondents who did not cooperate with HEIs are summarized in Table 5 below, together with the expected frequencies of the goodness of fit test.*

**Table 5.** Measured and expected response rates of the 3rd hypothesis.

|                | Measured | Expected |
| -------------- | -------- | -------- |
| Interested     | 15       | 11.5     |
| Not interested | 8        | 11.5     |
| TOTAL          | 23       | 23       |

The level of significance achieved (*p*-value 0.144) was greater than the chosen level of significance (0.05), so we will not reject the null hypothesis in favor of the alternative one. The hypothesis "More than half of the surveyed employers who did not cooperate with universities by offering internships to students would welcome cooperation in the future" could not be confirmed at the 5% level of significance.

*5.2. Questionnaire Survey of Inspectors of Regional Labor Inspectorates and State Labor Inspection Office of the Czech Republic*

The aim of the survey was to analyze current awareness of Industry 4.0, in particular whether inspectors actually encountered new technologies selected according to Section 4, during inspections in companies. The target group were inspectors stationed throughout the Czech Republic and based on their feedback it was also possible to obtain an updated figure of the territory already experiencing the implementation of Industry 4.0 technologies, also in view of revamping education and lifelong learning aspects. The survey was structured according to eight open questions and the total number of questionnaires evaluated was 86, out of which 35 inspectors had already encountered the implemented new technology.

Three additional working hypotheses were elaborated as follows:

4. *So far, the inspectors have not encountered implemented Industry 4.0 technology in an enterprise during an inspection.*
5. *From the inspector's point of view, the Czech legislation regarding new technologies is sufficient.*
6. *Inspectors are interested in Industry 4.0 professional training.*

**Hypothesis 4.** *The answers to question 4 were used to test this hypothesis: "Have you encountered implemented industry 4.0 technology in an enterprise during an inspection?" Response frequencies are summarized in Table 6.*

**Table 6.** Measured and expected response rates of the 4th hypothesis.

| Respondents    | Measured | Expected |
| -------------- | -------- | -------- |
| Encountered    | 35       | 43       |
| Not encountered| 51       | 43       |
| TOTAL          | 86       | 86       |

Level of significance achieved (*p*-value 0.084) was greater than the chosen level of significance (0.05), so we will not reject the null hypothesis in favor of the alternative. It is concluded that the explored hypothesis 4 could not be confirmed at a 5% level of significance.

**Hypothesis 5.** *The answers to the following question 6, "Are the laws in the Czech Republic regarding new technologies sufficient?" were used to test this hypothesis. The responses were summarized in Table 7.*

**Table 7.** Measured and expected response rates of the 5th hypothesis.

|  | Measured | Expected |
|---|---|---|
| Sufficient | 17 | 28.667 |
| Insufficient | 16 | 28.667 |
| I don't know | 53 | 28.667 |
| TOTAL | 86 | 86 |

The resulting level of significance (*p*-value 0.00001) was smaller than the chosen level of significance (0.05), so the null hypothesis is rejected in favor of the alternative one. The hypothesis "Are the laws in the Czech Republic regarding new technologies sufficient?" could be confirmed at a 5% significance level.

**Hypothesis 6.** *The answers to following question 8 were the basis to test this hypothesis: "Would you be interested in Industry 4.0 training?" The response numbers are summarized in Table 8.*

**Table 8.** Measured and expected response rates of the 6th hypothesis.

|  | Measured | Expected |
|---|---|---|
| Interested | 32 | 28.667 |
| Not interested | 18 | 28.667 |
| I don't know | 36 | 28.667 |
| TOTAL | 86 | 86 |

Since the *p*-value (0.044) was less than the statistical significant level level (0.05), the the tested null hypothesis is rejected in favor of the alternative one. The surveyed "Would you be interested in Industry 4.0 training?" is confirmed at a 5% significance level.

*5.3. Questionnaire Survey of Employees of Trade Unions of the Czech Republic*

The pilot questionnaire was devoted to the research of employees of trade unions working in the Czech Republic within the context of Industry 4.0. The aim of the questionnaire was to analyze current awareness of Industry 4.0, in particular whether trade union employees encountered new technologies during inspections in companies. The target group were the employees of trade unions stationed throughout the Czech Republic. It was their feedback that contributed to the mapping of the territory of the Czech Republic that has already seen the implementation of new Industry 4.0 technology for the future transformation of study fields in response to the 4th Industrial Revolution. The total number of evaluated questionnaires was 37, of which 30 employees of trade unions had already encountered the introduced new technology.

The questionnaire included 8 questions and three additional working hypotheses were prepared for this section of the study:

7. *Currently, the inspectors have not encountered implemented Industry 4.0 technology in an enterprise during an inspection.*
8. *From the perspective of a trade union employee, the Czech legislation is sufficient with regard to new technologies.*
9. *Trade union employees are interested in Industry 4.0 professional training.*

**Hypothesis 7.** *To test this hypothesis, the answers used to Question 4: "Have you encountered implemented industry 4.0 technology in an enterprise during an inspection?" The responses and expected numbers are summarized in Table 9.*

**Table 9.** Measured and expected responses of the 7th hypothesis.

|  | Measured | Expected |
|---|---|---|
| Encountered | 30 | 18.5 |
| Not encountered | 7 | 18.5 |
| TOTAL | 37 | 37 |

Since the achieved *p*-value (0.00016) was lower than the selected level of significance (0.05), we will reject the null hypothesis in favor of the alternative one. The hypothesis examined: "Did you encounter any Industry 4.0 technology during an inspection in a company?" could be confirmed at a 5% level of significance.

**Hypothesis 8.** *The answers to question 6 were used to test this hypothesis: "Are the laws in the Czech Republic regarding new technologies sufficient?" Response and expected numbers for the good fit test are summarized in Table 10.*

**Table 10.** Measured and expected responses of the 8th hypothesis.

|  | Measured | Expected |
|---|---|---|
| Sufficient | 5 | 12.333 |
| Not sufficient | 3 | 13.333 |
| I don't know | 29 | 12.333 |
| TOTAL | 37 | 37 |

The resulting level of significance (*p*-value = 0.00001) was smaller than the selecyted significance level (0.05), so we will reject the null hypothesis in favor of the alternative one. Explored hypothesis: "Is the legislation in the Czech Republic regarding new technologies sufficient?" could be confirmed at a 5% significance level.

**Hypothesis 9.** *The answers to question 8 were used to test the hypothesis, obtaining the results summarized in Table 11.*

**Table 11.** Measured and expected responses of the 9th hypothesis.

|  | Measured | Expected |
|---|---|---|
| Interested | 10 | 12.333 |
| Not interested | 4 | 13.333 |
| I don't know | 23 | 12.333 |
| TOTAL | 37 | 37 |

Since the resulting *p*-value (0.00048) was lower than the selected level of significance (0.05), the tested null hypothesis was rejected in favor of the alternative one. Explored hypothesis: "Would you welcome Industry 4.0 training?" was confirmed at 5% significance level.

*5.4. Student Questionnaire Survey*

The second questionnaire, focused on students' awareness of Industry 4.0, represents a very challenging item, as the acquisition of novel skills within the framework of Safety 4.0 can be complicated for an aging labor force not characterized by a minimal scholastic training [11]. In order to explore this topic, the selected target group consists of students of the Faculty of Safety Engineering, VSB-TUO. A total sample of 70 students participated in the questionnaire survey, as follows:

- 28 students of the first year of bachelor study;
- 15 students of the second year of the bachelor study of Process and Occupational Safety;
- 19 students of the third year of the bachelor study of Process and Occupational Safety;

- 8 students of the first year of master study in the field of Safety Engineering.

The questionnaire consisted of 5 questions with a maximum of 6 points. The success rate was set within the range of 4–6 points. A questionnaire with 3 or less points was evaluated as unsuccessful. The aim was not an individual assessment of the interviewees, but only a general overview of the level of basic knowledge of students.

Of the 70 students surveyed, only 27 passed the threshold of success. The remaining 43 were unsuccessful. The most successful year in the number of correct answers earning 5 points is the second year of bachelor study, followed by the 3rd year. The difference between 2nd and 3rd year is 22%. The first year of bachelor's study ranked a successful limit of 46%, which represents 4 correct answers to the questionnaire. The average number of correct answers is 3.1 for the whole course.

## 6. Discussion

As previously outlined, personnel and process safety issues should be regarded as highly relevant aspects in the world of sustainability, which should be considered according to an integrated vision avoiding unintended negative impacts (e.g., hazards to employers posed by improvements for environmental impact mitigation), or tension between goals (e.g., environment vs. environment). New digital technologies are affecting roles, processes and organizations, thus imposing changes in the society, government as well as in education. The safety culture is highly important in organizational settings, as it spurs the risk based thinking at all levels realizing the benefits of safety-sustainability nexus [22]. The concept of sustainability is complex and polyhedral so that its approach within the university context must be attentive to the needs of society and try to promote mutual exchange and collaboration [58]. As a follow-up of the statistical survey, in the following section we briefly outline a number of practical issues that could contribute to ensure a higher employability of graduates considering Industry 4.0 novel challenges.

Following the reasoning presented in [59], upon proper adaptation for the given explored domain as shown in Table 12, the effective modification of the learning scheme can be performed accounting for three main areas, namely:

- Knowing: the student engagement with relevant knowledge;
- Acting: the performative character in learning whereby capabilities are developed to take up knowledge and use it in new and challenging situations;
- Being: the required professional attitudes, dispositions and personal skills.

**Table 12.** The effective modification of the learning scheme.

| Areas | |
|---|---|
| Knowing | Modification of study plans of study program Fire Protection and Industrial Safety, Faculty of Safety Engineering, VSB-Technical University of Ostrava. |
| Acting | Increasing student awareness of Industry 4.0 in the form of lectures and seminars. Introduce intensive cooperation with the Czech Institute of Informatics, Robotics and Cybernetics (CIIRC), which is part of Czech Technical University in Prague. |
| Being | Approach companies throughout the Czech Republic and establish cooperation with universities to provide internships to students during their studies. |

Modification of study plans of study program Fire Protection and Industrial Safety, Faculty of Safety Engineering, VSB-TUO are proposed taking into account the previously outlined learning framework and trying to link individual courses in a coherent manner. As suggested for the interconnected sustainable development goals [60], specific and transversal competences need to be adequately developed.

In the first year of bachelor study, which is the same for all students, it is recommended to modify the subject Machines, Equipment, and Technology covering sensors and information technology and developing system thinking and system engineering approaches discussed in Section 2.

In the second, third, and fourth years of the bachelor's study program Safety of Work and Processes, it is recommended to modify the subjects Machine and Equipment Safety, Computer Science, Technical Safety, Occupational Safety Psychology, Risk Analysis and Hazards.

In the first and second years of the follow-up master's program in Safety Engineering, it is recommended to modify the subjects Diagnostics and Maintenance, Development Trends in Safety Engineering.

Just as an applicative example of additional learning abilities posed by advanced technologies reference is made to the hazards posed by collaborative robots (cobots). A cobot is more expensive and slower than a "normal" industrial robot, but for given industrial applications it seems preferable, even if it does not cooperate with humans, because the pressure information of its strain gauges (sensors) can be used. In this context, students shall adequately acquire an understanding of the following standards:

- ISO 13482: 2014 Robots and robotic devices—Safety requirements for personal care robots [61],
- ISO 15066: 2016 Robots and robotic devices—Collaborative robots [62].

In fact, from a practical point of view, the cobot can be viewed according to two perspectives:

1. The robot is in collaborative mode—in this case, ISO standard 15066: 2016 serves as guidance, the table of maximum forces (different forces for different body parts) that the robot can exert on human contact. Impact on the head and neck by any force is prohibited. If the robot meets these parameters, a person may approach to any proximity. If the robot does not meet these parameters, it is necessary to calculate the safety distance to stop.
2. Robot in non-collaborative mode—the minimum distance to the hazard zone must be properly calculated CSN EN ISO 10218-1: 2011 Robots and robotic devices—Safety requirements for industrial robots—Part 1: Robots.

When one comes to more practical oriented issues resulting from this study, it is suggested to consider three opportunities. The challenges summarized in the following are connected to the remark that the new process engineer generation will increasingly need IT skills, as their direct interactions with the plant will reduce and they will be required to increasingly interact with digital twin [2].

Firstly, to increase student awareness of Industry 4.0 in the form of lectures and seminars. This issue can also be included into other subjects of VSB-TUO faculties and, thus, increase students' awareness of the 4th Industrial Revolution implications and emphasize the importance of the study of this topic. Lectures and seminars can be led by professors and engineers from other universities, as well as by practitioners who own 4.0 technology. In general, it is recommended to involve more professionals, business owners, and employees in education. Their contribution in form of lectures, speech, and practical experience are very beneficial and motivating for students, as already experienced within University teaching curricula in different Countries.

Secondly, to introduce intensive cooperation and promote transdisciplinary learning and research approaches, e.g., with the Czech Institute of Informatics, Robotics and Cybernetics (CIIRC), which is part of Czech Technical University in Prague. CIIRC [63] is a research center and workplace in the field of industry, which is a highly respected institute in the Czech Republic not only in academic circles. It also serves students who attend it during their studies and can test practices and technologies that are already, or will be, implemented in the industry. This includes Testbed for Industry 4.0, which serves as a platform between business practitioners and academia. Here, it is certainly the golden rule that the most effective way is to learn from the best. The Testbed Ostrava [64] building for laboratory teaching called CPIT TL3 will soon be added to the modern university campus of VSB-TUO. The "FEI CPIT TL3 New Technology Platform" project was approved under the Research, Development, and Education Operational Program. The project focuses on improving the educational infrastructure of VSB-TUO through the addition of the CPIT

TL3 building itself and the acquisition of equipment to ensure quality education, especially in the new bachelor/master study programs. It is a comprehensive learning tested platform with Smart Factory (Digital Factory with Industry 4.0 Elements—IoT, Digital Twin, Big Data, Robotics, Predictive Maintenance, Virtual and Augmented Reality, Image Recognition and 3D Object Identification, Machine Learning), Home Care (flats for teaching new technologies of biomedical engineering) and Automotive (laboratories focusing on e-mobility, Functional Safety and Automotive Spice, simulation and testing of automotive electronic systems).

Thirdly, a close connection and involvement of industry personnel is considered a strategic challenge for delivering effective high-level learning outcomes and enhance students' attitudes and practical skills for entry into future professional practice. On these bases, an industrial network will be established throughout the Czech Republic process companies aiming at establishing close cooperation with universities and provide internships to students during their studies. This form of cooperation can be developed by both the university and by individual faculties, but also by students who can arrange cooperation near their home. They can select a company by sector, size, and mainly the subject of business. A challenging opportunity for companies to pay back their loyalty is to invite companies to the annual Career PLUS Job Fair, where industrial partners both improve networking and come into contact with students and academia. Additionally, a continuous effort for improving the courses over the coming years is considered a vital and essential part of the overall Safety 4.0 learning design, to be verified over time by students' feedback evaluation, starting from the benchmark exercise.

## 7. Conclusions

The overall concept of Industry 4.0 will affect not only the industry but also the safety, security, education, the legal framework, science and research as well as the global social system. Nowadays, it is necessary to start preparing the current generation for the new challenges and, thus, to develop human potential. This article addressed the current situation on the labor market from the perspective of employers and requirements for students, or graduates of Czech Republic universities.

Focusing on the hazards posed by new technologies, the need of a paradigm shift towards the here newly defined Safety 4.0 is highlighted. At least to our knowledge, this particular topic is explored for the first time in the scientific literature and the authors acknowledge that there are areas for improvement that can be further investigated. The target groups of the surveys and subsequent statistical evaluation were:

- companies whose respondents were employers of different industries,
- inspectors of the Regional Labor Inspectorates and the State Labor Inspection Office,
- employees of trade unions,
- students of the Faculty of Safety Engineering VSB-TUO.

The research survey was primarily interested in shedding light on the relationship between the company size and the economic and human resource investment on the risk management organization and process: results confirmed the critical importance of firm size on the running expenditure and actual people involvement in the overall risk management process and consequently expected safety levels.

Considering the preparedness of graduates for employment, the result was inadequate in terms of practical experience into the integration of OHS within Industry 4.0 and inconsistent in terms of training on emerging safety issues. It is important to ensure Safety 4.0 incorporation into chemical engineering education by a major effort to redesign academic programs. The here presented findings based on statistical surveys in Czech Republic may probably reflect similar situations in other European Countries currently experiencing Industry 4.0 revolution.

At last, the survey results suggested practical recommendations that can contribute to ensuring a higher learning value and skill development in view of the future graduate entry into Industry 4.0 professional practice, effectively facing Safety 4.0 challenges. However,

this study represents a first step limited to a small sample size and further evaluation will be required to test the effectiveness of the presented concepts.

**Author Contributions:** V.L.—Writing original draft, review and editing; K.S.—Survey design and statistics elaboration; B.F.—Conceptualization, review and supervision; A.B.—Conceptualization and supervision, review and editing. All authors have read and agreed to the published version of the manuscript.

**Funding:** This research received funding by Technology Agency of the Czech Republic under the project "Potential Impacts of Industry 4.0 on Operators 3.0 Jobs and Tertiary Education with Accordance of Safety Engineering", identification code TL01000470.

**Acknowledgments:** The authors gratefully acknowledge the valuable comments of anonymous reviewers.

**Conflicts of Interest:** The authors declare no conflict of interest.

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
