# Peer review of "Trends and Opportunities of Tertiary Education in Safety Engineering Moving towards Safety 4.0"

_sustainability, doi:10.3390/su13020524_

Round 1

Reviewer 1 Report

Dear Authors:

First of all, I want to congratulate you on the good job you have done. The research field they deal with is really interesting, and the contributions of their work to this field can be considered relevant to this area of knowledge.

Next, I provide some indications to improve the final paper.

Dear authors:

First of all, I want to congratulate you on the good job you have done. The research field they deal with is really interesting, and the contributions of their work to this field can be considered relevant to this area of knowledge.

Next, I provide some indications to improve the final paper.

El apartado 1, que es un apartado introductorio, se considera que está elaborado correctamente, bien redactado, con una estructura coherente, y apoyado adecuadamente en referencias correctas.

There is a possible error between the text on line 93-95, which indicates "A comprehensive example of immediate readability regarding the paradigm of industry 93 4.0 evolution is provided by Keller [13] illustrating by simple numerical indicators the increasing 94 importance of digitalization comparing their relevance in 2018 and 2050 ", and figure 1, which refers to the year 2025. Which of the two is correct? It is recommended to clarify this question.

On the other hand, the effort to clarify the why and wath of Safety 4.0 is positively appreciated, to adequately frame the study.

In line 146 below, it is recommended to support in a bibliographic reference the relationship between safety and sustainability that is established in the text and is exposed in the figure.

The methodological proposal is considered adequate for the purposes of the research approach.

In the presentation of results, it is recommended to specify if the questions used have been prepared by the researchers, have been validated, or have been taken from other previous existing studies.

Discussion and conclusions are considered appropriate.

Due to the nature of the research journal "Sustainability", it is recommended to introduce a paragraph in which the conclusions that link safety and sustainability are especially highlighted.

The references section is considered correct. In this field of knowledge, the studies by Zamora-Polo, Sánchez-Martín, published in Sustainability (https://www.mdpi.com/2071-1050/11/15/4224) (https: // www.mdpi.com/2071-1050/11/13/3533) If they consider it appropriate, they can take them into account as possible references.

I hope that these contributions contribute to the improvement of the final result of the paper.

Best regards

Reviewer 2 Report

The paper can be published.

This manuscript is a resubmission of an earlier submission. The following is a list of the peer review reports and author responses from that submission.

Round 1

Reviewer 1 Report

The research is very interesting and statistically clearly written.

Author Response

Dear reviewer,

Thank you for your positive evaluation, there are no comments, so we prepare the revision version mainly accounting on the reports by the second and third reviewer.

Thank you for your effort and encouraging words certainly stimulating our research. In this regard, we added acknowledgments to all reviewers in the manuscript.

Ales Bernatik

Corresponding co-author

Reviewer 2 Report

Dear authors:

First of all, I would like to congratulate you on the work you have done. it is interesting and brings knowledge to the field of knowledge they deal with.

Author Response

Dear reviewer,

Thank you for your positive evaluation and encouraging words. In the following we append a list of changes as reaction to your comments. In this regard, we added formal acknowledgments to all reviewers at the end of the manuscript.

Thank you very much for the time spent and the several useful comments.

Ales Bernatik

Corresponding co-author

Reviewer 3 Report

The following work tries to study the occupational risks of Industry 4.0 and the analysis of the current situation on the labor market from the perspective of employers and requirements for students, or graduates of Czech Republic universities.

In my opinion, this work has serious limitations. I comment on these deficiencies below:

1.- Introduction section: (i) this section does not follow a clear and defined common thread; (ii) the literature review is poor and unstructured; (iii) the context and main objective are not clearly defined because of the above aspects. Authors should define (for example safety 4.0 or new and emerging risk) and clearly distinguish new and emerging risks associated with Industry 4.0 from the analysis of the current situation on the labor market. Perhaps this double aspect of this paper adds confusion; (iv) the structure of section 1.1 should be justified; (v) section 1.2 is confusing. This section does not explain the entire manuscript. Lines 280-283 looks like a result and is therefore confusing.

2.- Materials and methods: this section is clearly insufficient. This section does not describe the methodology followed in the manuscript (only a small part).

3.- Results: the results presented are not understandable within the framework of a well-defined methodology, because of the above limitations. In any case, in this section, there are methodological aspects that should be part of the Materials and methods section.

4.- Discussion: this section is not understandable within the framework of the methodology and results presented.

5.- Conclusions: the conclusions are very poor since they are essentially a summary of the manuscript. The authors point out only one limitation of this work, which, indeed, is important.

Finally, I recommend that the authors make a strong effort to restructure and improve this manuscript. To do this, first, a review of the literature on new and emerging risks and Industry 4.0 clearer, orderly, and updated is necessary. Second, an adequate description of the methodology used is very important. Third, the results should be presented in accordance with this methodology, in a clear and well-ordered manner. Finally, the discussion of the results and their conclusions should be understood according to the methodology and results presented.

Author Response

Dear reviewer,

we appreciated very much all comments from reviewers. It is always that one is confronted with possible improvements to present a more complete argumentation, serious omissions, words that were not intended for the meaning that others attach to them, questions raised but no answers provided, simple typos etc. In this regard, we added acknowledgments to all reviewers in the revised manuscript.

Here we append a list of changes as reaction to your comments, combined with other received suggestions.

At last, thank you again for your encouraging words and the time spent for reviewing and practical suggestions, allowing us to improve the paper. We consider these as a reward for the many hours to collect the material and writing the paper.

Ales Bernatik

Corresponding co-author

Round 2

Reviewer 2 Report

Dear authors.
Congratulations on the improvements made to your paper.
Best regards

Reviewer 3 Report

In my opinion, this revised version has essentially the same problems as the previous version. These problems are methodological and are transferred to the whole of the manuscript. I exposed these limitations in my previous report (I recommended that the authors make a strong effort to restructure and improve this manuscript).